# Effectiveness of a Visual Interactive Game on Oral Hygiene Knowledge, Practices, and Clinical Parameters among Adolescents: A Randomized Controlled Trial

**DOI:** 10.3390/children9121828

**Published:** 2022-11-26

**Authors:** Ram Surath Kumar K, Apurva Prashant Deshpande, Anil V. Ankola, Roopali M. Sankeshwari, Sagar Jalihal, Vinuta Hampiholi, Atrey J. Pai Khot, Mamata Hebbal, Sree Lalita Kotha, Lokesh Kumar S

**Affiliations:** 1Department of Public Health Dentistry, KAHER’s KLE Vishwanath Katti Institute of Dental Sciences, JNMC Campus, Nehru Nagar, Belagavi 590010, Karnataka, India; 2Department of Periodontics, KAHER’s KLE Vishwanath Katti Institute of Dental Sciences, JNMC Campus, Nehru Nagar, Belagavi 590010, Karnataka, India; 3Department of Preventive Dental Sciences, College of Dentistry, Princess Nourah bint Abdulrahman University, P.O. Box 84428, Riyadh 11671, Saudi Arabia; 4Department of Basic Dental Sciences, College of Dentistry, Princess Nourah bint Abdulrahman University, P.O. Box 84428, Riyadh 11671, Saudi Arabia; 5Department of Oral Medicine and Radiology, KAHER’s KLE Vishwanath Katti Institute of Dental Sciences, JNMC Campus, Nehru Nagar, Belagavi 590010, Karnataka, India

**Keywords:** game, experimental, health education, health knowledge, oral health, oral hygiene, attitudes, practice

## Abstract

This study aimed to evaluate the effect of a novel interactive game-based visual performance technique (IGVP) and conventional oral health educational (OHE) talk on plaque control, gingival health, and oral hygiene knowledge and practices in 12–15-year-old schoolchildren. A single-blinded randomized controlled trial was undertaken in a private primary school in Belagavi, Karnataka, India. A total of 100 children aged 12–15 years were randomly assigned to either a conventional OHE talk (control group, *n* = 50) or the IGVP technique (test group, *n* = 50), using a computer-generated table of random numbers. A self-designed, pre-validated closed-ended questionnaire was collected from both groups, followed by clinical examination carried out using gingival and plaque index, pre- and post-intervention, at three months follow-up. There was a significant reduction in the mean gingival score and plaque score in the test group after intervention, indicating a 58.7% and 63.4% reduction, when compared to the control group, which had a 2.8% and 0.7% reduction, respectively (*p* < 0.001). The test group showed a significant increase in the percentage of knowledge gained (22.4%), compared to control group (7.8%). The IGVP technique proved to be more effective than a conventional OHE talk in the reduction of the plaque score, gingival score, and in the improvement of the knowledge of oral hygiene maintenance and its application.

## 1. Introduction

Oral health is a key element of general health [1]. Strong oral health practices are believed to be important when instilled in young minds in order to achieve positive lifelong oral health [2]. Setting up oral health education (OHE) programs in schools is important to implement adaptive changes and improve oral health. School days are the formative stages for a child, to transform both physically and mentally into a promising adult [3]. School is the most suitable habitat for the implementation of health education programmes, because it serves as a platform to encourage healthy self-care practices in a large group of children. The education imparted must inspire the recipient’s mind and not just fill their heads with facts. Education is a threefold process of imparting knowledge, developing skills and interests, and developing attitudes and life values [4]. One way to achieve the above goals is to integrate education and entertainment, thus making the learning process enjoyable [5]. The present study incorporated this blend to create the interactive game-based visual performance (IGVP) technique, used in the present study, to help gain knowledge and reinforce practices among 12–15-year-old schoolchildren. It has been found that conventional OHE, on its own, does not give satisfactory results in changing oral health behaviour and attitude [6]. However, due to the dearth of supporting literature, the same cannot be said about IGVP, an innovative technique employed in this study to deliver OHE.

Games and cartoon animations can be incorporated as teaching tools to create visual awareness, improve attention span, and aid with memory strategies and reasoning in a fun and innovative manner [7,8]. Kahoot, an online interactive game-based learning platform, is one of the best examples of gamified learning sessions that enhance educational practices with new technological capabilities. It was developed to accommodate learners of all ages and languages and make learning interesting [9]. Hence, the aim of this study was to evaluate the effects of a novel validated IGVP technique vs. the conventional OHE talk on plaque control, gingival health, and oral hygiene knowledge and practices in 12–15-year-old schoolchildren pre-and post-intervention.

### 1.1. Null Hypothesis

There is no difference in plaque control, gingival health, and oral hygiene knowledge and practices when imparted with conventional oral health talk and IGVP technique among the 12–15-year-old schoolchildren at the end of three months.

### 1.2. Alternative Hypothesis

There is a difference in plaque control, gingival health, and oral hygiene knowledge and practices when imparted with conventional oral health talk and IGVP technique among the 12–15-year-old schoolchildren at the end of three months.

## 2. Materials and Methods

### 2.1. Study Design and Study Setting

The study was carried out as a double arm single-blinded randomized controlled type in a private primary school in Belagavi, Karnataka, India, during the period from February to April 2022. The trial was registered under the Clinical Trials Registry—India with the CTRI number CTRI/2022/02/040252. Consolidated Standards of Reporting Trials (CONSORT) guidelines were being followed.

### 2.2. Sample Size Calculation

The sample size for the study was estimated using the GPower program (G*Power Version 3.1.9.4 statistical software). The sample size was estimated to be 40 children in each group, accounting for a total sample size of 80 at a power of 0.80 with a 0.05 alpha error [10]. Hence, to increase the statistical power and take into consideration lost participants during follow-up, it was decided to enroll 100 children in total, with 50 children in each group.

### 2.3. Inclusion Criteria

Children between the ages of twelve and fifteen years.

### 2.4. Exclusion Criteria

Children and their parents who did not give assent and consent to participate in the study.

Children with underlying systemic disease and/or other special health care needs.

### 2.5. Ethical Considerations

The ethical clearance was obtained from the Institutional Research and Ethics Committee (IRB number: EC/NEW/2021/2435) with the reference number: 1510, dated: 28/11/2021. This study strictly adhered to the ethical standards of human experimentation and the Helsinki Declaration of 1975, as revised in 2000.

### 2.6. Pilot Study

A self-designed 17-item closed-ended questionnaire was prepared and assessed for reliability, using Cronbach’s alpha (0.85), and validity, using the content validity ratio (0.82). A pilot study was conducted for a group of 15 children between the ages of 12–15 years. This was carried out to check the feasibility of the study and the response of the children to the visual aids. Children requested a reduction in the speed of the visual content to perceive it better, and the adjustment was carried out based on their feedback. The results of the pilot study were not included in the main study.

### 2.7. Assessment Plan

The assessment was carried out in three phases.

#### 2.7.1. Preparatory Phase

Investigators were pre-calibrated for recording the gingival and plaque index and were overseen by subject experts. Intra-examiner and inter-examiner reliability were calculated (0.82, 0.86) and (0.83, 0.89), respectively, using kappa statistics, which indicated a substantial level of agreement.

#### 2.7.2. Randomization and OHE Intervention

A total of 100 volunteering children were included to participate in the study. A simple random sampling technique was employed by using a lottery method. Children were randomly assigned into groups to receive either a conventional oral health talk (control group, *n* = 50) or the IGVP technique (test group, *n* = 50), using a computer-generated table of random numbers. Allocation concealment was completed using the SNOSE (sequentially numbered, opaque, sealed envelope) technique (Figure 1). Questionnaires in both English and the regional language (Kannada) were distributed, and the responses, such as sociodemographic details and information on oral hygiene knowledge and practices, were collected by the investigators. This was followed by a clinical examination by two investigators, who were blinded about the group allocation. They recorded gingival and plaque scores, using the Loe and Silness gingival index [11], and Silness and Loe plaque index [12]. After the baseline assessment, another investigator delivered the conventional OHE talk and IGVP technique to the control group and the test group children, respectively. Figure 1 shows the study design.

Conventional oral health education technique: After the clinical examination, children of the control group attended a 15 min health education talk given by a professionally trained investigator who was proficient in educating children. The health education talk was inclusive of an introduction to oral health and its significance in general health, the five golden rules for maintaining effective oral health, and finally, the brushing technique (Modified Bass Technique) was demonstrated.

Interactive game-based visual performance technique (IGVP): (a) Visual—A customized OHE animated video of 5.94 min duration, with subtitles in English, was played on the screen for the test group. The content displayed an introduction to oral health and its significance in general health (2.06 min), the brushing technique (2.30 min), and the five golden rules for maintaining effective oral health (1.58 min). (b) Game—In the second step of the study, a 30 min online game-based question and answer (Q&A) interactive session using the Kahoot application, in context with the OHE animated video, was conducted. (c) Performance—A demonstration of the brushing technique (Modified Bass Technique) on the tooth model was given by the investigator. The children were individually trained to mimic the brushing technique on the tooth model to perfect and perform easily on their own. In the tooth brushing training, every child was asked to perform brushing on the tooth model until they were perfect and could perform with ease, spending approximately 4 min with each child.

#### 2.7.3. Follow-Up

The final assessment of the gingival and plaque scores of all the children in the test and control groups was conducted three months after the initial assessment, using the same indices. The investigators’ bias was minimized by carrying out an examination by the same investigators who had been blinded during the grouping of children. Oral hygiene knowledge and practices were reassessed after three months to estimate the impact of the OHE techniques among children. The indices were recorded in the morning (between 10:00 am and 1:00 pm) during the period of after breakfast but before lunch with no snack time in between. The control group also received a similar type of OHE (IGVP technique) after the completion of phase three.

The questionnaire consisted of 17 items, of which 13 were knowledge-based, to be answered by the 50 children of each group and were scored ranging from 0 to 13. The correct response was scored as “1” and the incorrect response as “0”. The total knowledge score was computed based on the response of each child. The overall score was totalled by a simple sum of responses.

### 2.8. Statistical Analysis

Data obtained were entered into Microsoft Excel (2020) and analyzed using the SPSS^®^ (IBM Corp. Released 2012 IBM SPSS Statistics for Windows, Version 21.0. Armonk, NY, USA: IBM Corp.). In the statistical evaluation, the normality of the data distribution was determined using the Shapiro–Wilk test, and the data were found to be normally distributed. The descriptive statistics were presented as mean ± standard deviation for continuous variables and as frequencies with percentages for categorical variables. The following univariate analyses were used: the McNemar and Chi-square analysis to assess the differences in response to pre- and post-OHE intervention in both groups, and paired and unpaired *t*-tests to compare the mean plaque score, gingival score, and knowledge score in and between the conventional OHE and IGVP techniques, respectively. Drop-out analysis was performed with *t*-tests [13]. The statistical significance was set at *p* ≤ 0.05.

## 3. Results

The demographic characteristics of the study population of conventional OHE technique and IGVP technique are presented in Table 1.

Figure 2 summarizes the mean gingival and plaque scores in the control group and test group, pre- and post-OHE intervention. An unpaired *t*-test revealed no statistically significant difference between the groups at the baseline, with the mean gingival score (*p* = 0.775) and plaque score (*p* = 0.426). In the test group, a paired *t*-test revealed a greater reduction in the mean gingival and plaque scores from the baseline to three months interval (*p* < 0.001). Figure 3 shows a greater reduction in the mean gingival score and plaque score in the test group after intervention, indicating a 58.7% and 63.4% reduction when compared to the control group, which had a 2.8% and 0.7% reduction, respectively (*p* < 0.001).

Table 2 and Table 3 summarize the pre- and post-OHE intervention oral hygiene knowledge and practices, respectively. McNemar test showed that the percentage of correct answers was significantly increased after the OHE intervention (*p* < 0.05). An unpaired *t*-test revealed the baseline knowledge score in both groups was almost equal (control group: 7.30 ± 1.31; test group: 7.29 ± 1.72) and statistically insignificant (*p* = 0.192). The mean knowledge score post-OHE intervention of the control group (7.88 ± 1.97) and test group (8.94 ± 1.27) indicated a statistically significant difference between the groups, *p* < 0.001 (Figure 2). Following the OHE intervention, the test group showed a significant increase in the percentage of knowledge gained (22.4%) when compared to the control group (7.8%) using a paired *t*-test, *p* < 0.001 (Figure 3). Nonetheless, the dropout analysis ensured that the present study was adequately powered, dropouts and completers did not differ significantly, and differential attrition biases did not exist.

## 4. Discussion

OHE for schoolchildren is a widely adopted primary strategy for the prevention of oral diseases, owing to its cost-effectiveness and simplicity in administration [14,15]. Studies have demonstrated that cartoon-animated audiovisual aids and game-based education implemented for schoolchildren are powerful tools for educating and providing thought-stimulating methods of learning [16,17]. The use of fun-based learning interventions (connect the dots games [1,18], crosswords and quizzes [3], flashcards [1], snakes and ladders [19,20], augmented reality board games [21], video games [22,23], smartphone oral hygiene applications [24], and animated audiovisual resources [25]) have been suggested to be a perfect motivating tool to encourage children to take part in self-care activities. The use of gaming in OHE can be a good alternative for educating basic health concepts, thereby, strengthening cognitive development and confidence [26]. One such interactive gaming session, Kahoot (https://kahoot.com (accessed on 01 February 2022)), an online quiz platform that enables educators to create an entertaining gaming environment, is used in the present study [27].

This interventional study aims to compare the effect of conventional OHE talk and IGVP technique on plaque control, gingival health, and oral hygiene knowledge and practices, among 12–15-year-old schoolchildren in Belagavi city. This study is a first-of-a-kind to employ the IGVP technique to create a positive influence on tooth brushing behaviour and habits to improve knowledge and practices of oral health maintenance and compare it to the conventional OHE talk.

### 4.1. Comparison of Gingival and Plaque Scores

The mean gingival and plaque scores of both groups at the baseline were similar. However, in the post-OHE intervention, the results of the IGVP group revealed a significant reduction in the mean gingival and plaque scores at three months when compared to the baseline. This may be attributed to the use of animated videos along with question and answer based interactive gaming sessions that have allowed the children to comprehend the topics better and adopt them into their routine oral hygiene habits. The lack of improvement in oral hygiene in the control group was probably due to children not implementing what they had learnt through conventional health talk. The decrease in gingival and plaque scores and improved oral hygiene measures in the IGVP group compared to the control group can be attributed to the knowledge gained in the interventional phase. Improved plaque scores were sustained over a three month period, proving that the effects of the IGVP learning technique persisted over time. This finding agreed with studies conducted by Kashyap et al. [10] and Malik et al. [3], which showed that implementing crossword and quiz game-based OHE programmes resulted in a significant decrease in plaque scores. Similarly, Aljafari et al. [28] reported that implementing video game-based OHE programmes resulted in a significant decrease in plaque scores.

### 4.2. Comparison of Oral Hygiene Knowledge and Practices

A self-designed questionnaire was used to assess the knowledge and oral hygiene practices of children who perceived oral and general health as two distinct unrelated elements. However, in post-OHE intervention, the children gained knowledge that oral and general health are interrelated. Questions pertaining to the tooth cleaning agent, brushing time, tooth cleaning habits, mouth rinsing after meals, sugar-containing sticky food consumption, and periodic visits to the dentist elicited correct responses from children in both groups at the baseline. This pre-existing basic knowledge regarding oral hygiene maintenance and its awareness at the baseline can be attributed to the fact that it was conveyed in the primary classes. Children are also exposed to advertisements and the showcasing of oral health products. The above reasons may have elicited correct responses to the questions. However, most of the children were unaware of the frequency of toothbrushing, periodic replacement of toothbrushes, regular dental visits, and the correct brushing technique. The post-IGVP assessment had a higher number of correct responses. This indicates that children may have understood the relative importance of oral health to general health, the brushing technique, and the five golden rules for maintaining effective oral health. These correct responses point towards a favourable impact of the IGVP technique on children in the test group.

In the present study, baseline knowledge scores in both groups were almost equal. However, significant improvement in knowledge following the OHE was seen in the IGVP group when compared to children in the control group. Such an interactive game, along with the visual aid, may have had a better impact on these children and generated interest, resulting in a better understanding and a gain in knowledge. There was a significant increase in the knowledge scores in the IGVP group compared to the control group. These findings were in accordance with previous studies by Kashyap et al. [10], which showed that game-based education (i.e., crosswords and quiz with a PowerPoint presentation) proved very effective in improving oral health knowledge. Our findings are in accordance with those of Sharma et al. [19], Malik et al. [3], and Maheswari et al. [20], who used game-based OHE to improve knowledge, health behaviour, and practices. The studies by Sinor et al. [8] and Anwar et al. [25] also reported that a cartoon-animated audiovisual aid significantly promoted knowledge regarding oral hygiene maintenance, in comparison to those who attended the conventional oral health talk. Both techniques improved the practice behaviour of children following the OHE, which proves that the conventional OHE along with demonstration of brushing technique did provide good results in the control group. However, a significant improvement was seen in the IGVP group. The findings agreed with the study conducted by GeethaPriya et al., which reported that game-based education (i.e., modified snakes and ladders board game) was as effective as the flashcard method in improving the oral health behaviour and practices of schoolchildren [29]. Furthermore, Aljafari et al. reported that using an OHE video game improved dietary knowledge, toothbrushing, and dietary practices in children [28].

Mukhi et al. reported that both the cartoon video animation method and game method (i.e., connect the dots) were effective in improving the oral hygiene knowledge and attitude as well as the oral hygiene status of schoolchildren [18]. Systematic reviews by Nakao et al. [30] and Gauthier et al. [31] also supported the utilization of game-based health education (i.e., board games) being beneficial in imparting knowledge for behavioural modifications, such as the promotion of healthy eating, smoking cessation, etc. The consistency within the results and the existing literature supports the hypothesis that the IGVP technique proved to be more effective in the reduction of gingival and plaque scores, as well as in the improvement of knowledge and its application in oral hygiene maintenance when compared to a conventional OHE talk. For all the above reasons, the null hypothesis is rejected. The present study revealed that the incorporation of the IGVP technique for children in the school environment had a positive influence on tooth brushing behaviour and habits, leading to a progressive improvement in knowledge and its application in the maintenance of oral hygiene. The favourable impact of this technique on the oral health status of schoolchildren may serve as a road map for conducting this research on a countrywide platform. However, the use of technology in educating children has its inherent problems, such as:Access to potentially harmful platforms by children.Children who depend on games are often socially inept in real-world interactions.Computer and other electronic device use could result in health hazards, such as eye strain and other physical issues.The technologies required for active participation might be rather expensive, resulting in a gap between students who have access to them and those who do not [32].

### 4.3. Limitations

The study was conducted with a limited follow-up period.

### 4.4. Future Prospects and Recommendations

To substantiate the results of this study, longitudinal studies, incorporating numerous health education sessions engaging children, teachers, and parents with follow-up at various intervals to assess knowledge retention after the termination of health education, are required. Such studies can be extrapolated to children of all age groups, different socioeconomic status, and schools in different geographical locations.

### 4.5. Clinical Significance

Specific OHE methods in suitable populations have a long-lasting and deep-rooted impact on our society when the IGVP technique is delivered with suitable cognitive, psychological, and psychosocial dimensions in a child-friendly environment. Hence, this method can be used as an integral tool to increase awareness among children.

## 5. Conclusions

The IGVP technique proved to be more effective than the conventional OHE talk in the reduction of gingival and plaque scores, as well as in the improvement of knowledge and its application in oral hygiene maintenance. The incorporation of the IGVP technique, following the provision of a health education programme for schoolchildren, was an easy, fun-loving, engaging, and cost-effective method.

## Figures and Tables

**Figure 1 children-09-01828-f001:**
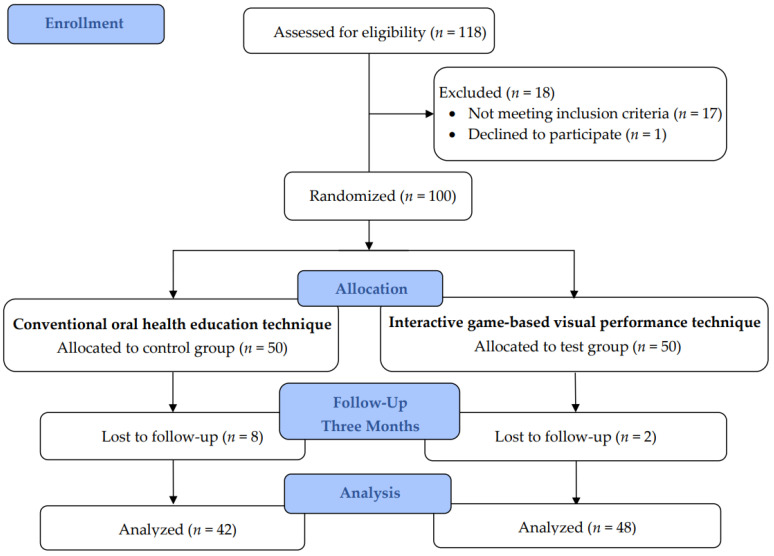
Consolidated Standards of Reporting Trials (CONSORT) diagram.

**Figure 2 children-09-01828-f002:**
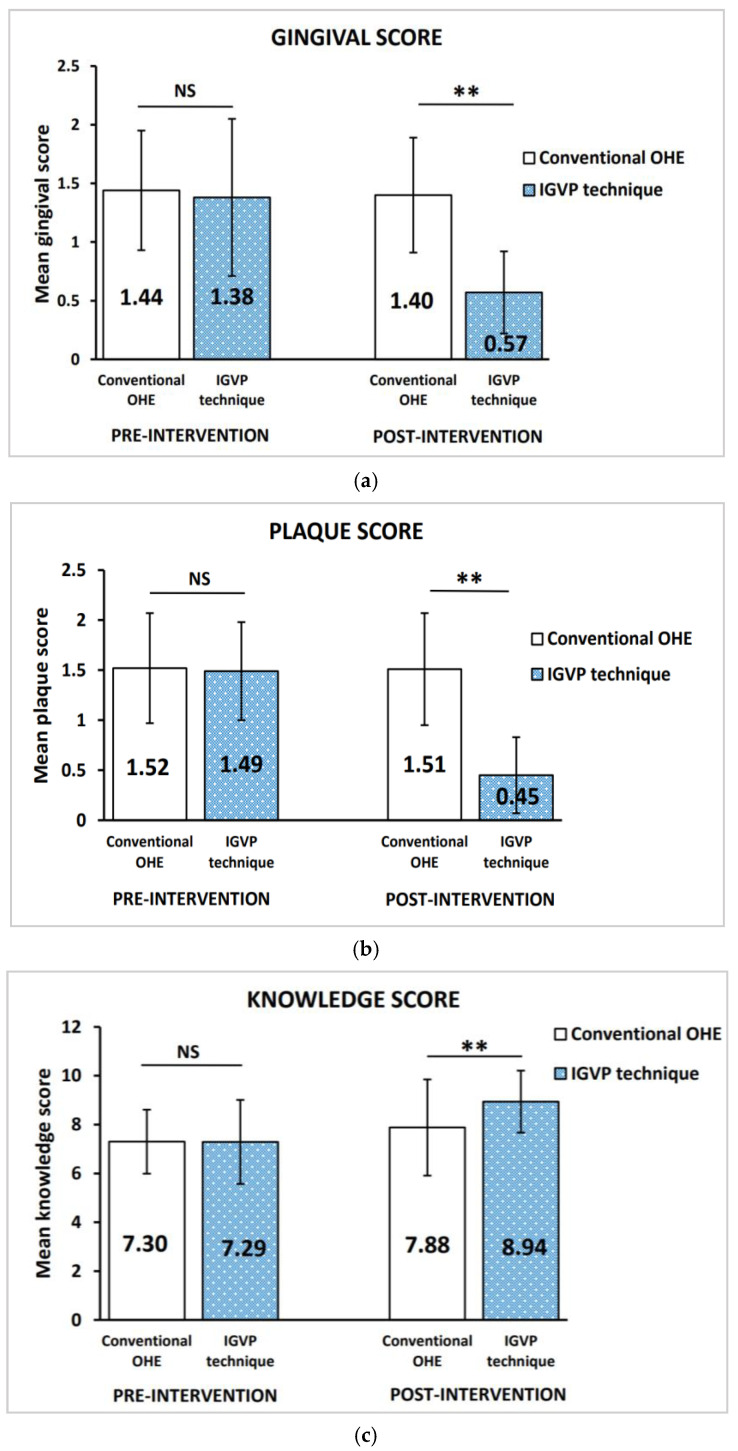
Comparison of the (**a**) mean gingival score, (**b**) plaque score, and (**c**) knowledge score in conventional OHE group and IGVP technique group. All values are expressed as mean ± standard deviation. The statistical test used: Unpaired *t*-test; ** Statistically significant, *p* ≤ 0.001; NS: not significant.

**Figure 3 children-09-01828-f003:**
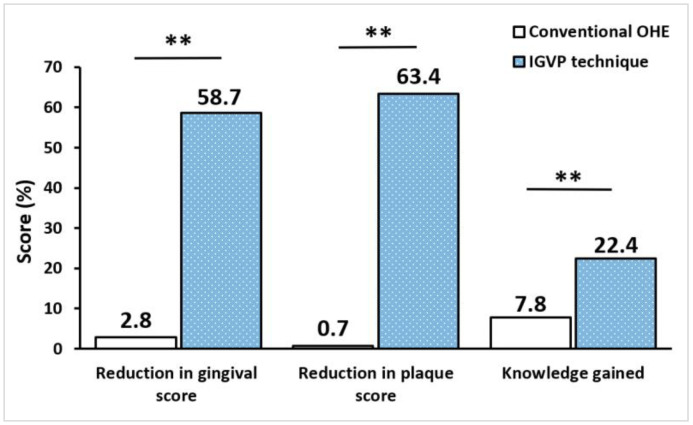
Percentage reduction in the mean gingival score, plaque score, and knowledge gained in the conventional OHE group and IGVP technique group. All values are expressed as percentages. The statistical test used: Unpaired *t*-test; ** Statistically significant, *p* ≤ 0.001.

**Table 1 children-09-01828-t001:** Demographic characteristics of the study population of conventional OHE technique and IGVP technique.

Demographic Characteristics		Conventional OHE (*n* = 42)	IGVP Technique (*n* = 48)
Age	Mean age ± SD	13.92 ± 1.47	13.35 ± 1.22
Gender	Male	25 (59.5%)	27 (56.3%)
	Female	17 (40.5%)	21 (43.8%)
Socioeconomic status	Lower middle class	38 (90.5%)	42 (87.5%)
	Upper lower class	4 (9.5%)	6 (12.5%)

IGVP: Interactive game-based visual performance; OHE: Oral health education. All values are expressed as frequency with percentages (in parentheses).

**Table 2 children-09-01828-t002:** Comparison of oral hygiene knowledge in the study population of the conventional OHE technique and IGVP technique before and after the intervention.

Questions		Response Frequencies *n* (%)						
		Response	Conventional OHE (*n* = 42)		Statistics	IGVP Technique (*n* = 48)		Statistics
			Pre	Post	*p* Value	Pre	Post	*p* Value
1.	Does oral hygiene have any role in the maintenance of general health?	Yes ^α^	34 (81%)	40 (95.2%)	0.109	36 (75%)	47 (97.9%)	<0.001 **
		No ^β^	8 (19%)	2 (4.8%)		12 (25%)	1 (2.1%)	
2.	Which among the two is the best tooth cleaning agent?	Toothpaste ^α^	35 (83.3%)	40 (95.2%)	0.125	43 (89.6%)	47 (97.9%)	0.219
		Toothpowder ^β^	7 (16.7%)	2 (4.8%)		5 (10.4%)	1 (2.1%)	
3.	How long should you brush your teeth?	2 to 3 minutes ^α^	22 (52.4%)	26 (61.9%)	0.388	14 (29.2%)	30 (62.5%)	0.007 *
		45 seconds ^β^	20 (47.6%)	16 (38.1%)		34 (70.8%)	18 (37.5%)	
4.	How many times should you brush your teeth in a day?	Twice a day ^α^	16 (38.1%)	40 (95.2%)	<0.001 **	4 (8.3%)	28 (58.3%)	<0.001 **
		Once in a day ^β^ Once in two days ^β^	26 (61.9%)	2 (4.8%)		44 (91.7%)	20 (41.7%)	
5.	Which is the best time to brush your teeth?	Morning and night ^α^	32 (76.2%)	34 (81%)	0.791	28 (58.3%)	30 (62.5%)	0.791
		Only morning ^β^ Only night ^β^	10 (23.8%)	8 (19%)		20 (41.7%)	18 (37.5%)	
6.	How do you clean your teeth regularly?	Brushing with toothpaste, toothbrush and rinsing after meal ^α^	24 (57.1%)	36 (87.2%)	0.023 *	28 (58.3%)	40 (83.3%)	0.023 *
		Brushing with finger ^β^ Brushing with toothpaste and toothbrush ^β^	18 (42.9%)	6 (14.3%)		20 (41.7%)	8 (16.7%)	
7.	When do you have to change your toothbrush?	Three months once ^α^	10 (23.8%)	14 (33.3%)	0.289	20 (41.7%)	44 (91.7%)	<0.001 **
		One month once ^β^ Two months once ^β^ Six months once ^β^	32 (76.2%)	28 (66.7%)		28 (58.3%)	4 (8.3%)	
8.	Which brushing technique is the best to clean your teeth?	Gentle downward and upward strokes along with circular motion ^α^	10 (23.8%)	38 (90.5%)	<0.001 **	26 (54.2%)	44 (91.7%)	<0.001 **
		Horizontal scrub motion ^β^	32 (76.2%)	4 (9.5%)		22 (45.8%)	4 (8.3%)	
9.	What will happen if you do not brush your teeth regularly?	Both of the above ^α^	34 (81%)	40 (95.2%)	0.109	32 (66.7%)	40 (83.3%)	0.077
		Tooth decay causing tooth loss ^β^ Gum disease causing tooth loss ^β^ I don’t know ^β^	8 (19%)	2(4.8%)		16 (33.3%)	8 (16.7%)	
10.	You should clean your teeth after every meal.	True ^α^	39 (92.9%)	41(97.6%)	0.625	40 (83.3%)	46 (95.8%)	0.109
		False ^β^	3 (7.1%)	1(2.4%)		8 (16.7%)	2 (4.2%)	
11.	Sweet and sticky foods containing sugar are healthy for your teeth.	False ^α^	30 (71.4%)	34 (81%)	0.454	26 (54.2%)	42 (87.5%)	<0.001 **
		True ^β^	12 (28.6%)	8 (19%)		22 (45.8%)	6 (12.5%)	
12.	Periodic check-up visits to a dentist are important to maintain the health of your mouth.	True ^α^	34 (81%)	36 (85.7%)	0.754	40 (83.3%)	42 (87.5%)	0.791
		False ^β^	8 (19%)	6 (14.3%)		8 (16.7%)	6 (12.5%)	
13.	How often should you visit the dentist?	Once in 6 months ^α^	4 (9.5%)	12 (28.6%)	0.039 *	10 (20.8%)	14 (29.2%)	0.454
		Once in a month ^β^ Once in 3 months ^β^ Once in a year ^β^	38 (90.5%)	30 (71.4%)		38 (79.2%)	34 (70.8%)	

IGVP technique: Interactive game-based visual performance technique. All values are expressed as a frequency with percentages (in parentheses); ^α^ denotes correct response, and ^β^ denotes wrong response. The statistical test used: McNemar test; Level of significance: * *p* ≤ 0.05 is considered statistically significant, ** *p* ≤ 0.001 is considered a highly statistically significant association.

**Table 3 children-09-01828-t003:** Comparison of oral hygiene practices in the study population of the conventional OHE technique and IGVP technique before and after the intervention.

Questions		Response Frequencies *n* (%)						
		Response	Conventional OHE (*n* = 42)		Statistics	IGVP Technique (*n* = 48)		Statistics
			Pre	Post	*p* Value	Pre	Post	*p* Value
1.	Do you use fluoride-containing toothpaste?	Yes ^α^	36 (85.7%)	40 (95.2%)	0.114 ^§^	34 (70.8%)	47 (97.9%)	<0.001 ^§,^**
		No ^β^	6 (14.3%)	2 (4.8%)		14 (29.2%)	1 (2.1%)	
2.	Do you use dental floss?	Yes ^α^	16 (38.1%)	20 (47.6%)	0.125 ^§^	10 (20.8%)	40 (83.3%)	<0.001 ^§,^**
		No ^β^	26 (61.9%)	22 (52.4%)		38 (79.2%)	8 (16.7%)	
3.	Do you clean your tongue?	Yes ^α^	18 (42.9%)	23 (54.8%)	0.063 ^§^	14 (29.2%)	42 (87.5%)	<0.001 ^§,^**
		No ^β^	24 (57.1%)	19 (45.2%)		34 (70.8%)	6 (12.5%)	
4.	Where did you get this information?	Parents/guardians	30 (71.4%)	26 (61.9%)	<0.001 ^||,^**	30 (62.5%)	6 (12.5%)	<0.001 ^||,^**
		Teachers	2 (4.8%)	2 (4.8%)		16 (33.3%)	0 (0%)	
		Dentist	10 (23.8%)	4 (9.5%)		2 (4.2%)	0 (0%)	
		Have not received any information on this (pre-intervention)	0 (0%)	-		0 (0%)	-	
		From this health education module (post-intervention)	-	10 (23.8%)		-	42 (87.5%)	

IGVP: Interactive game-based visual performance; OHE: Oral health education. All values are expressed as a frequency with percentages (in parentheses); ^α^ denotes good practice, and ^β^ denotes bad practice. The statistical test used: ^§^ McNemar test and ^||^ Chi-square test; Level of significance: ** *p* ≤ 0.001 is considered a highly statistically significant association.

## Data Availability

Data available on request to maintain confidentiality. The data presented in this study are available on request from the PI (First author). The data are not publicly available due to detailed information about the participants present in the data.

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
