# Peer review of "Effectiveness of a Visual Interactive Game on Oral Hygiene Knowledge, Practices, and Clinical Parameters among Adolescents: A Randomized Controlled Trial"

_children, 2022, doi:10.3390/children9121828_

Round 1
Reviewer 1 Report
Title: Too long
The authors are suggested to concise the title
Abstract: Accurate
Key words: use Mesh terms
Introduction:
Rephrase the below sentence: "It is believed that strong oral health practices are important from a young age to achieve positive lifelong oral health [2]" to "Strong oral health practices are believed to be important from a young age to achieve positive lifelong oral health"
Line No 41: remove "int terms of"
Lien No 42: remove "thereby"
Line No 49 and 51: "One of the ways of achieving the above goals is to integrate education and entertainment, thus making the process of learning an enjoyable one [5]" rewrite as mentioned below.
"One way to achieve the above goals is to integrate education and entertainment, thus making the learning process enjoyable [5]."
The author should explain the gap in the literature and the need for the present study.
The introduction should drive to the purpose of the study.
"Research hypothesis" should be like "hypothesis"
State Null hypothesis and alternative hypothesis.
Methods:
Sample calculation (2.2)
The content has taken from "Teaching Preschool Children Correct Toothbrushing Habits Through Playful Learning Interventions: A Randomized Controlled Trail ( 10.1016/j.pedn.2020.08.001)
The sample size was calculated based on power analysis and the authors stated that sample size is a limitation.
Inclusion and exclusion criteria are clear and rewrite with all necessary valid points
Pilot study:
Who was involved in the pilot study
What was their experience?
Randomization:
kindly explain the simple random sampling technique
Language:
The authors used English and the regional language (Kannada)
The questionnaire was translated to the local language, retranslated to English, and checked the missing content.
The experience of investigators involved in the study.
Both the investigators have seen all patients or distributed equal samples.
If separately seen calibration should be done under the supervision of an sr academician.
lines 119 and 120: remove the Authors initials.
The video was in which language
Private primary school (median of instruction as not mentioned) since authors stated in the Kannada language raises the issue.
Why follow-up was in three months?
What is the reference for three months follow-up?
Who monitored the children during the visit?
Is there any evaluation that happened during one month follow-up?
Stasts:
1. More detail needs to be given as indicated below.
2. This section should be based on a plan of conducting the analyses in the order: frequencies, univariate methods, multivariable methods.
Results:
The comparison of the base line, one month and three months should add additional value.
In the conventional group, a surprising fall in good oral hygiene practices this fall will be a very good point to discuss.
The authors stated Knowledge and practices, according to the KAP model
What are the measures used for knowledge scoring?
What is population's knowledge status on Oral hygiene practice?
These are very important key factors in the study.
Discussion:
This should not repeat detailed findings from the Results.
The focus should be on describing in words the key messages from this survey in the context of the current literature.
Discuss reasons why the findings reported differ from or confirm other work. What has been added to what is already known? ".
The discussion part should contain a comparison of your study findings with existing literature, however, the authors just stated a few of the findings from prior studies.
Lines: 279 -283: Supporting statements are missing
Language issues should be carefully checked, discussion part there a lot of definitive statements are present.
Lines 262-289: remove authors' initials.
Limitations:
there are a lot of issues to discuss in limitations.
Based on G power authors found the sample is 40 with attrition authors have chosen 50. I could not see that sample size is a potential limitation.
Clinical significance:
Maximum content looks similar to
Sharma et al 2001........."Comparison between Conventional, Game-based, and Selfmade Storybook-based Oral Health Education on Children’s Oral Hygiene Status: A Prospective Cohort Study
Conclusion:
Should be based on your aims and objectives
Moreover, the conclusion looks like a general
References:
Approprite
Author Response
Dear reviewer:
Greetings of the Day
This is regarding the manuscript entitled " Effectiveness of Interactive Game-Based Visual Performance Technique on Plaque Control, Gingival Health, Oral Hygiene Knowledge and Practices of 12- to 15-Year-Old School Children: A Randomized Controlled Trial" (Manuscript ID: children-1961235) which was sent for revision from the Journal name: Children. We, truly acknowledge and are indebted for the detailed review of the paper and the comments received from your end. We value the suggestions which help accentuate the quality of the research paper.
All the corrections have been made in the revised article. The Reviewer’s comments as well as the authors' replies to the comments have been attached.
Kindly consider it for further evaluation.
Thank you.
Reply to the reviewer’s comments
|
1. |
Title: Too long The authors are suggested to concise the title |
|
Reply: The title has been modified. “Effectiveness of Interactive Game-Based Visual Performance Technique on Plaque Control, Gingival Health, Oral Hygiene Knowledge and Practices among School Children: A Randomized Controlled Trial” |
|
|
2. |
Abstract: Accurate |
|
3. |
Key words: use Mesh terms |
|
Reply: Mesh terms have been included. |
|
|
4. |
Introduction: Rephrase the below sentence: "It is believed that strong oral health practices are important from a young age to achieve positive lifelong oral health [2]" to "Strong oral health practices are believed to be important from a young age to achieve positive lifelong oral health" |
|
Reply: Has been rephrased The same changes have been applied in the revised manuscript (line numbers: 40-42). |
|
|
5. |
Line No 47: remove "int terms of" |
|
Reply: Has been removed and rephrased The same changes have been applied in the revised manuscript (line numbers: 42-43). |
|
|
6. |
Lien No 42: remove "thereby" |
|
Reply: Has been removed The same changes have been applied in the revised manuscript (line number: 43). |
|
|
7. |
Line No 49 and 51: "One of the ways of achieving the above goals is to integrate education and entertainment, thus making the process of learning an enjoyable one [5]" rewrite as mentioned below. "One way to achieve the above goals is to integrate education and entertainment, thus making the learning process enjoyable [5]." |
|
Reply: Has been rephrased. The same changes have been applied in the revised manuscript (line number: 49-50). |
|
|
8. |
The author should explain the gap in the literature and the need for the present study. The introduction should drive to the purpose of the study. |
|
Reply: Has been added Line numbers: 51-57 “The present study incorporated this blend to create this interactive game-based visual performance (IGVP) technique used in the present study to help gain knowledge and reinforce practices among 12- to 15-year-old school children. It has been found that the conventional OHE, on its own, does not give satisfactory results in changing oral health behaviour and attitude [6]. However, due to the dearth of supporting literature, the same cannot be said about IGVP, an innovative technique employed in the study to deliver OHE” |
|
|
9. |
"Research hypothesis" should be like "hypothesis" State Null hypothesis and alternative hypothesis. |
|
Reply: Has been added The same changes have been applied in the revised manuscript (line numbers: 67-74). “ 1.1.Null Hypothesis: There is no difference in plaque control, gingival health, oral hygiene knowledge and practices when imparted with conventional oral health talk and IGVP technique among the 12–15-year-old school children at the end of three months. 1.2.Alternative Hypothesis: There is a difference in plaque control, gingival health, oral hygiene knowledge and practices when imparted with conventional oral health talk and IGVP technique among the 12–15-year-old school children at the end of three months.” |
|
|
10. |
Methods: Sample calculation (2.2) The content has taken from "Teaching Preschool Children Correct Toothbrushing Habits Through Playful Learning Interventions: A Randomized Controlled Trail ( 10.1016/j.pedn.2020.08.001) The sample size was calculated based on power analysis and the authors stated that sample size is a limitation. |
|
Reply: Has been modified. The same changes have been applied in the revised manuscript (line numbers: 82-88). “The sample size for the study was estimated using the GPower program (G*Power Version 3.1.9.4 statistical software). The sample size was estimated to be 40 children in each group accounting for a total sample size of 80 at a power of 0.80 with 0.05 alpha error [10]. Hence, to increase statistical power and take into consideration of lost participants during follow-up, it was decided to enroll 100 children in total, with 50 children in each group”. |
|
|
11 |
Inclusion and exclusion criteria are clear and rewrite with all necessary valid points |
|
Reply: Has been changed. The same changes have been applied in the revised manuscript (line numbers: 89-94). “2.3. Inclusion Criteria: Children between the ages of twelve and fifteen years. 2.4. Exclusion Criteria: Children and their parents who did not give assent and consent to participate in the study. Children with underlying systemic disease and/or other special health care needs”
|
|
|
12 |
Pilot study: Who was involved in the pilot study What was their experience? |
|
Reply: “A self-designed 17-item closed-ended questionnaire was prepared and assessed for reliability using Cronbach's alpha (0.85), and validity using the content validity ratio (0.82). A pilot study was conducted for a group of 15 children between the ages of 12- to 15-years. This was carried out to check the feasibility of the study and the response of the children to the visual aids. The results of the pilot study were not included in the main study.”
Children requested for a reduction in the speed of the visual content to perceive it better and the adjustment was carried out based on their feedback. The same changes have been applied in the revised manuscript (line numbers: 100-106). |
|
|
13 |
Randomization: kindly explain the simple random sampling technique |
|
Reply: “A total of 100 volunteering children were included to participate in the study. A simple random sampling technique was employed by using lottery method.” The same has been added in the revised manuscript (line numbers: 117-118). |
|
|
14 |
Language: The authors used English and the regional language (Kannada) The questionnaire was translated to the local language, retranslated to English, and checked the missing content. |
|
Reply: The questionnaire was administered in both languages to the children depending on their ease of communication. The questionnaire was translated to the local language (Kannada) and retranslated to English to check for missing contexts and validate the translation by language experts. |
|
|
15 |
The experience of investigators involved in the study. |
|
Reply: The experience of investigators involved in the study: Investigator 1 – Four years of experience. Investigator 2 – Three years of experience. |
|
|
16 |
Both the investigators have seen all patients or distributed equal samples. |
|
Reply: Yes, both investigators have examined the children by distributing the samples equally. |
|
|
17 |
If separately seen calibration should be done under the supervision of an sr academician. |
|
Reply: Investigators were pre-calibrated for recording gingival and plaque index and were overseen by subject experts. (Line numbers: 110-111) |
|
|
18 |
lines 119 and 120: remove the Authors initials. |
|
Reply: The Authors initials have been removed. The changes have been applied in the revised manuscript. |
|
|
19 |
The video was in which language |
|
Reply: The video was in the English language |
|
|
20 |
Private primary school (median of instruction as not mentioned) since authors stated in the Kannada language raises the issue. |
|
Reply: Median of instruction was in English |
|
|
21 |
Why follow-up was in three months? |
|
Reply: In order to evaluate the effectiveness and knowledge retention of the IGVP technique incorporated in oral hygiene knowledge and practices. The difference in plaque and gingival scores can be assessed after a tenure of three months and above of plaque buildup. |
|
|
22 |
What is the reference for three months follow-up? |
|
Reply: Maheswari UN, Asokan S, Asokan S, Kumaran ST. Effects of conventional vs. game based oral health education on children’s oral health related knowledge and oral hygiene status – A prospective study. Oral Health Prev Dent. 2014; 12:331 6. https://doi.org/10.3290/j.ohpd.a32677 Aljafari A, Gallagher JE, Hosey MT. Can oral health education be delivered to high‐caries‐risk children and their parents using a computer game?–A randomised controlled trial. International journal of paediatric dentistry. 2017; 27(6):476-85. https://doi.org/10.1111/ipd.12286
|
|
|
23 |
Who monitored the children during the visit? |
|
Reply: The assisting clerk with investigators. |
|
|
24 |
Is there any evaluation that happened during one month follow-up? |
|
Reply: No evaluation has been conducted during one month follow-up. |
|
|
25 |
Stasts: 2. This section should be based on a plan of conducting the analyses in the order: frequencies, univariate methods, multivariable methods. |
|
Reply: Has been added The same changes have been applied in the revised manuscript (line numbers: 169-174). However multivariate analysis was not used in this study. |
|
|
26 |
Results: The comparison of the baseline, one month and three months should add additional value. |
|
Reply: There was no follow-up after one month from the baseline assessment in the present study. The detailed comparison between baseline and three months was presented in Tables 2 and 3; Figures 2 and 3. |
|
|
27 |
In the conventional group, a surprising fall in good oral hygiene practices this fall will be a very good point to discuss. |
|
Reply: Line numbers. 254-256 “The lack of improvement in oral hygiene of the control group was probably due to children not implementing what they had learnt through conventional health talk.” |
|
|
28 |
The authors stated Knowledge and practices, according to the KAP model What are the measures used for knowledge scoring? |
|
Reply: Line numbers. 161-163 “The total knowledge score was computed based on the response of every child. Each negative response was scored as “0” and positive as “1”. The overall score was a simple sum of responses ranging from 1 to 13.” |
|
|
29 |
What is population's knowledge status on Oral hygiene practice? These are very important key factors in the study. |
|
Reply: Line numbers. 199-200 “the baseline knowledge score in both groups was almost equal (control group: 7.30 ± 1.31; test group: 7.29 ± 1.72) and statistically insignificant (p = 0.192).” |
|
|
30 |
Discussion: This should not repeat detailed findings from the Results. The focus should be on describing in words the key messages from this survey in the context of the current literature. Discuss reasons why the findings reported differ from or confirm other work. What has been added to what is already known? ". The discussion part should contain a comparison of your study findings with existing literature, however, the authors just stated a few of the findings from prior studies. |
|
Reply: Has been modified. The same changes have been applied in the revised manuscript (line numbers: 258-262, 296-305). |
|
|
31 |
Lines: 279 -283: Supporting statements are missing |
|
Reply: Has been added. The same changes have been applied in the revised manuscript (line numbers: 300-306). |
|
|
32 |
Language issues should be carefully checked, discussion part there a lot of definitive statements are present. |
|
Reply: Has been checked and corrected. |
|
|
33 |
Lines 262-289: remove authors' initials. |
|
Reply: Authors' initials have been removed. The same changes have been applied throughout the revised manuscript. |
|
|
34 |
Limitations: There are a lot of issues to discuss in limitations. Based on G power authors found the sample is 40 with attrition authors have chosen 50. I could not see that sample size is a potential limitation. |
|
Reply: Has been modified. The same changes have been applied in the revised manuscript (line number: 326). |
|
|
35 |
Clinical significance: Maximum content looks similar to Sharma et al 2001........."Comparison between Conventional, Game-based, and Selfmade Storybook-based Oral Health Education on Children’s Oral Hygiene Status: A Prospective Cohort Study |
|
Reply: Compelled to keep the medical terminologies but has now been modified. The same changes have been applied in the revised manuscript (line numbers: 333-337). “The study was conducted with a limited follow-up period.” |
|
|
36 |
Conclusion: Should be based on your aims and objectives Moreover, the conclusion looks like a general
|
|
Reply: Has been corrected. The same changes have been applied in the revised manuscript (line numbers: 338-343). “IGVP technique proved to be more effective than the conventional OHE talk in the reduction of gingival and plaque scores as well as in the improvement of knowledge and its application in oral hygiene maintenance. The incorporation of IGVP technique following the provision of a health education program in school children was an easy, fun-loving, engaging and cost-effective method.” |
|
|
37 |
References: Appropriate |

Reviewer 2 Report
Dear Authors,
This paper addresses an interesting topic, well designed with well prepared manuscript, however, I would recommend some modifications before considering its publication. Below these are some suggestions for You:
-
Authors:
1.A. All authors should use institutional emails
-
Material and methods:
2.A. Who performed statistical analysis?
-
Results:
3.A. Figure 2. I suggest converting the arrangement of the A-C charts from horizontal to vertical
3.B. Table 1. Answers: Some data appears to be missing (or rather layout is broken - denoted wrong responses should be much more visible they are taken into account as an one answer)
Best regards and good luck
Author Response
Dear reviewer:
Greetings of the Day
This is regarding the manuscript entitled " Effectiveness of Interactive Game-Based Visual Performance Technique on Plaque Control, Gingival Health, Oral Hygiene Knowledge and Practices of 12- to 15-Year-Old School Children: A Randomized Controlled Trial" (Manuscript ID: children-1961235) which was sent for revision from the Journal name: Children. We, truly acknowledge and are indebted for the detailed review of the paper and the comments received from your end. We value the suggestions which help accentuate the quality of the research paper.
All the corrections have been made in the revised article. The Reviewer’s comments as well as the authors' replies to the comments have been attached.
Kindly consider it for further evaluation.
Thank you.
Reply to the reviewer’s comments
|
1. |
Authors: 1. A. All authors should use institutional emails |
|
Reply: Institutional email IDs have been added. Few authors did not have institutional mail. |
|
|
2. |
Material and methods: 2. A. Who performed statistical analysis? |
|
Reply: Authors (A.P.K., M.H.) who have been extensively trained in biostatistics performed analysis. |
|
|
3. |
Results: 3. A. Figure 2. I suggest converting the arrangement of the A-C charts from horizontal to vertical |
|
Reply: Has been modified (Figure 2a-c). |
|
|
3.B. Table 1. Answers: Some data appears to be missing (or rather layout is broken - denoted wrong responses should be much more visible they are taken into account as one answer) |
|
|
Reply: Thank you for bring it to notice, in order to increase the visibility and clarity in the presentation of data of wrong responses (Table 2 - Q4, Q5, Q6, Q7, Q9, Q13), authors have made clear demarcation between the correct and multiple wrong. |

Reviewer 3 Report
Dear authors,
Thank you for submitting the interesting manuscript entitled "Effectiveness of Interactive Game-Based Visual Performance Technique on Plaque Control, Gingival Health, Oral Hygiene Knowledge and Practices of 12- to 15-Year-Old School Children: A Randomized Controlled Trial". I have some questions and suggestions to address before considering it for publication.
Please, submit your manuscript for an English proofreading service.
Introduction:
The introduction is poorly written and needs improvements before considering for publication. Also, include the study's aim in the introduction.
Research hypothesis = Hypothesis 0 (H0)
Materials and Methods:
Sample Size Calculation
Please, explain the method used to calculate the sample size and include the "drop-off" number in the text.
Ethical Considerations:
Include the IRB number.
Results:
Include a Social-demographic table from the subjects (CONSORT)
Conclusion:
After including the study aim, please, review your conclusion.
Regards,
#Reviewer
Author Response
Dear reviewer:
Greetings of the Day
This is regarding the manuscript entitled " Effectiveness of Interactive Game-Based Visual Performance Technique on Plaque Control, Gingival Health, Oral Hygiene Knowledge and Practices of 12- to 15-Year-Old School Children: A Randomized Controlled Trial" (Manuscript ID: children-1961235) which was sent for revision from the Journal name: Children. We, truly acknowledge and are indebted for the detailed review of the paper and the comments received from your end. We value the suggestions which help accentuate the quality of the research paper.
All the corrections have been made in the revised article. The Reviewer’s comments as well as the authors' replies to the comments have been attached.
Kindly consider it for further evaluation.
Thank you.
Reply to the reviewer’s comments
|
1. |
Introduction: The introduction is poorly written and needs improvements before considering for publication. Also, include the study's aim in the introduction. |
|
Reply: Thank you for your valuable suggestion. In response to the reviewer’s comments, we authors have made the necessary corrections. We have corrected the grammatical errors as much as possible. The aim of the study has been included in the revised manuscript (line numbers: 63-66) “the aim of this study was to evaluate the effect of a novel validated IGVP technique and conventional OHE talk on plaque control, gingival health, oral hygiene knowledge and practices in 12–15-year-old school children pre-and post-intervention.” |
|
|
2. |
Research hypothesis = Hypothesis 0 (H0) |
|
Reply: Has been added (Line numbers: 67-74) |
|
|
3. |
Materials and Methods: Sample Size Calculation Please, explain the method used to calculate the sample size and include the "drop-off" number in the text. |
|
Reply: “The sample size for the study was estimated using the GPower program (G*Power Version 3.1.9.4 statistical software). The sample size was estimated to be 40 children in each group accounting for a total sample size of 80 at a power of 0.80 with 0.05 alpha error [10]. Hence, to increase statistical power and take into consideration of lost participants during follow-up, it was decided to enroll 100 children in total, with 50 children in each group.” (Line numbers: 82-88) |
|
|
4. |
Ethical Considerations: Include the IRB number. |
|
Reply: (IRB number: EC/NEW/2021/2435) Has been included in the revised manuscript. (Line numbers: 97, 354) |
|
|
5. |
Results: Include a Social-demographic table from the subjects (CONSORT). |
|
Reply: Has been included (Table 1). |
|
|
6. |
Conclusion: After including the study aim, please, review your conclusion. |
|
Reply: Has been reviewed. “IGVP technique proved to be more effective than the conventional OHE talk in the reduction of gingival and plaque scores as well as in the improvement of knowledge and its application in oral hygiene maintenance. The incorporation of IGVP technique fol-lowing the provision of a health education program in school children was an easy, fun-loving, engaging and cost-effective method.” (Line numbers: 338-343) |

Reviewer 4 Report
I would like to thank the authors for this interesting, informative, and well written manuscript that evaluated the effectiveness of Interactive Game-Based Visual Performance Technique on Plaque Control, Gingival Health, Oral Hygiene Knowledge, and Practices of 12- to 15-Year-Old School Children.
Some suggestions for clarifications are listed below.
Introduction:
11- P2 L 59-63: (Interactive 59 game-based visual performance (IGVP) technique, a novel oral health educational (OHE) 60 program in the form of an animated visual playback with subtitles in English was used, 61 which included a game-based question and answer interaction using Kahoot, followed by 62 the demonstration of the brushing technique on a tooth model.) This part looks like methodology done by the investigators, please remove from introduction section.
Methods:
11- The authors mentioned at their inclusion criteria that children between 12-15 years were selected. Please justify why was this adolescent age group particularly selected?
22- Please mention whether results of subjects from the pilot study were included in the main study or not.
33- P4 L 119: (followed by clinical examination by two investigators who were blinded about the group). Please mention when this clinical examination was carried out at baseline and follow up ! Before or after snack time? Do you think examination time had positively or negatively affected the plaque score recorded ?
Discussion:
1- This section is very well written; however, it would be appreciated if the authors mentioned whether they accepted or rejected their research hypothesis.
Author Response
Dear reviewer:
Greetings of the Day
This is regarding the manuscript entitled " Effectiveness of Interactive Game-Based Visual Performance Technique on Plaque Control, Gingival Health, Oral Hygiene Knowledge and Practices of 12- to 15-Year-Old School Children: A Randomized Controlled Trial" (Manuscript ID: children-1961235) which was sent for revision from the Journal name: Children. We, truly acknowledge and are indebted for the detailed review of the paper and the comments received from your end. We value the suggestions which help accentuate the quality of the research paper.
All the corrections have been made in the revised article. The Reviewer’s comments as well as the authors' replies to the comments have been attached.
Kindly consider it for further evaluation.
Thank you.
Reply to the reviewer’s comments
|
Introduction: |
|
|
|
P2 L 59-63: (Interactive 59 game-based visual performance (IGVP) technique, a novel oral health educational (OHE) 60 program in the form of an animated visual playback with subtitles in English was used, 61 which included a game-based question and answer interaction using Kahoot, followed by 62 the demonstration of the brushing technique on a tooth model.) This part looks like methodology done by the investigators, please remove from introduction section. |
|
Reply: Has been removed |
|
|
Methods: |
|
|
1. |
The authors mentioned at their inclusion criteria that children between 12-15 years were selected. Please justify why was this adolescent age group particularly selected? |
|
Reply: 1. Children below 12 years may have difficulties in comprehending the technological aspects (Kahoot online game) and adhering to the instructions. 2. Recording of gingival and plaque scores requires specific fully erupted permanent tooth. 3. Mixed dentition has more of debris accumulation due to varying stages of tooth eruption. 4. Selection of 12-15 years age group was based WHO index age group classification. |
|
|
2. |
Please mention whether results of subjects from the pilot study were included in the main study or not. |
|
Reply: The results of subjects from the pilot study were not included in the main study. Has been included in the revised manuscript (Line numbers: 105-106). |
|
|
3. |
P4 L 119: (followed by clinical examination by two investigators who were blinded about the group). Please mention when this clinical examination was carried out at baseline and follow up ! Before or after snack time? Do you think examination time had positively or negatively affected the plaque score recorded ? |
|
Reply: Thank you for your question, indices were recorded in the morning (between 10:00 am and 1:00 pm) during the period of after breakfast and before lunch with no snack time in between. (Line numbers: 156-158) |
|
|
Discussion: |
|
|
|
This section is very well written; however, it would be appreciated if the authors mentioned whether they accepted or rejected their research hypothesis. |
|
Reply: Thank you for your valuable suggestion. Rejection of null hypothesis has been added in the revised manuscript. “The consistency within the results and the existing literature support the hypothesis that the IGVP technique proved to be more effective in the reduction of gingival and plaque scores as well as in the improvement of knowledge and its application in oral hygiene maintenance when compared to conventional OHE talk. For all the above rea-sons, the null hypothesis is rejected.” (Line numbers: 306-310). |
|

Round 2
Reviewer 1 Report
Dear Authors
The manuscript better than old version.
However, all the suggested changes weee not made
There are still methodological issues.
the results was not explained properly.
discussion need updated citations.
Round 3
Reviewer 1 Report
The authors responded well to all the comments
Still yet to answer these questions with clarity:
The comparison of the base line, one month and three months should add additional value.
In the conventional group, a surprising fall in good oral hygiene practices this fall will be a very good point to discuss.
The authors stated Knowledge and practices, according to the KAP model
What are the measures used for knowledge scoring?
What is the population's knowledge status on Oral hygiene practice?
Who was involved in the pilot study
What was their experience?
Randomization:
kindly explain the simple random sampling technique
